# Non-invasive current collectors for improved current-density distribution during $CO_2$ electrolysis on super-hydrophobic electrodes

Hugo-Pieter Iglesias van Montfort [1], Mengran Li[1,2], Erdem Irtem [1], Maryam Abdinejad [1], Yuming Wu[3], Santosh K. Pal[1], Mark Sassenburg[1], Davide Ripepi [1], Siddhartha Subramanian[1], Jasper Biemolt[1], Thomas E. Rufford [3] & Thomas Burdyny [1] ✉

Electrochemical reduction of $CO_2$ presents an attractive way to store renewable energy in chemical bonds in a potentially carbon-neutral way. However, the available electrolyzers suffer from intrinsic problems, like flooding and salt accumulation, that must be overcome to industrialize the technology. To mitigate flooding and salt precipitation issues, researchers have used super-hydrophobic electrodes based on either expanded polytetrafluoroethylene (ePTFE) gas-diffusion layers (GDL's), or carbon-based GDL's with added PTFE. While the PTFE backbone is highly resistant to flooding, the non-conductive nature of PTFE means that without additional current collection the catalyst layer itself is responsible for electron-dispersion, which penalizes system efficiency and stability. In this work, we present operando results that illustrate that the current distribution and electrical potential distribution is far from a uniform distribution in thin catalyst layers (~50 nm) deposited onto ePTFE GDL's. We then compare the effects of thicker catalyst layers (~500 nm) and a newly developed non-invasive current collector (NICC). The NICC can maintain more uniform current distributions with 10-fold thinner catalyst layers while improving stability towards ethylene ($\geq$ 30%) by approximately two-fold.

The electrochemical reduction of carbon dioxide ($CO_2$RR) has been gaining traction as a means of storing renewable energy in sustainable fuels and chemicals like carbon monoxide, ethylene, and ethanol. As a result, research and development efforts are shifting from understanding fundamental reaction mechanisms towards industrial scale-up and practical challenges of electrochemical conversion processes[1,2]. Gas diffusion electrodes (GDEs) are now widely used to overcome mass-transport limitations at the cathode, where $CO_2$ is reduced, to perform $CO_2$RR at industrially relevant reaction rates, to yield value-added carbon products[3,4].

The electrochemical reduction of $CO_2$ using GDEs is however challenged by several problems that curb the upscaling of this technology to large industrial applications (Fig. 1a). First, the competing hydrogen evolution reaction (HER) forces $CO_2$ electrolysis towards alkaline environments, where HER has a higher overpotential than $CO_2$RR. This shift to alkaline environments, in turn, causes $CO_2$ to

[1]Department of Chemical Engineering, Delft University of Technology; 9 van der Maasweg, Delft 2629HZ, the Netherlands. [2]Department of Chemical Engineering, The University of Melbourne, Parkville, VIC 3010, Australia. [3]School of Chemical Engineering, The University of Queensland, St. Lucia, QLD 4072, Australia. ✉e-mail: t.e.burdyny@tudelft.nl

**Fig. 1 | The local environment characteristics at an ePTFE and flooded carbon GDLs. a** Advantages and disadvantages of super-hydrophobic ePTFE and carbon-based GDE architectures for CO₂ electrolysis. **b** Close-up sketch of an ePTFE electrode, current collection sketched in blue. **c** Sketch of a carbon-based GDE, current collection sketched in blue. **d** Modeled local $CO_{2,aq}$ concentration at steady-state close to the catalyst layer (coral) in the ePTFE electrode at −300 mA cm⁻². **e** Modeled local $CO_{2,aq}$ concentration at steady-state close to the catalyst layer (coral) for the carbon GDE, for non-flooded and fully flooded cases at −300 mA cm⁻².

from ePTFE with 250–500 nm fibrils stretched from 1 to 10 μm nodes, whereas the BL is a polymeric support material formed by coarse laminates of polyethylene or polypropylene (PE, PP). In contrast to carbon-based GDE's which can quickly lose their ability to prevent flooding[9,16,17], non-conducting PTFE GDE's maintain hydrophobicity when exposed to sustained electrochemical potentials and moderate hydrostatic pressures. With further modifications, like covering the catalyst layer with ionomers, electrolyzers using ePTFE type GDE's have then shown the ability to deliver both high stability (>60 h) and selectivity (>80% FE at 1.17 A cm⁻²) towards multicarbon ($C_{2+}$) products during CO₂RR (Fig. 1a)[18].

However, a caveat of the excellent non-wetting and CO₂ diffusion properties of ePTFE GDE's is the non-conductive nature of the polymeric backbone. Upon electron flow from the anode to the cathode, electrons must then be conducted transversely through the catalyst layer rather than the GDL (Fig. 1b), which is contrary to the conductive backbone of carbon-based GDE's (Fig. 1c). As catalyst layer thicknesses (0.1–5 μm) are much thinner than carbon-GDE's (200–500 μm)[19], current distribution through the catalyst layer results in much higher in-plane ohmic resistances than in carbon-GDE's, which deter the industrial scalability of electrolyzers with ePTFE GDE's. Even for a 5 cm² electrode area, large current density disparities would occur due to the increased in-plane resistance to electron-conduction, negatively impacting product selectivity towards CO₂RR. When the current distribution is constrained in a poorly conducting electrode, it can be expected that catalytic activity is concentrated close to the current collector, while the rest of the electrode surface operates at reduced potential/current density or is even inactive for CO₂RR—a problem that only worsens for increasing geometric catalyst areas. This current distribution disparity on the CO₂RR electrodes (ePTFE and carbon-GDE's) has, to our knowledge, never been studied in detail before.

In order to design and assess efficient electrode architecture designs for CO₂RR, we need to better understand the operando behavior of the electrode—even more so considering that local activity (and, thus, local overpotential) can have a critical influence on the product distribution observed[20–23]. If current distribution is not uniform due to high conductivity resistance in the catalyst layer, the potential applied to the catalyst will vary spatially, jeopardizing the long-term stability and product selectivity. We have shown previously that local heat production (probed by temperature sensing of the electrode's GDL) is a valid and accurate proxy for an electrode's activity[24]. Infrared (IR) thermography then provides a basis of understanding local conditions and electrode behavior during operation of a CO₂ electrolyzer.

Herein, we use infrared thermography to demonstrate the potential problems of current distribution in ePTFE-based electrodes with thin catalyst layers in a state-of-the-art flow cell CO₂RR electrolyzer. Using a 1D reaction-diffusion model of the gas-liquid interfaces in the ePTFE and carbon-based electrodes, we show how the GDE structure and operational stability affects the local availability of CO₂ in the catalyst layer and the $C_{2+}$ product selectivity. Then we analyze the current distribution in ePTFE electrodes and examine the deterioration of the thin catalyst layers deposited on the expanded PTFE-layer. Infrared thermographs display poor current density distribution for 50 nm catalyst layers, where the active region is taxed with a current load around 5 times higher than the average. Finally, we showcase a non-invasive current collector (NICC) as an alternative catalyst layer design to improve the current collection and distribution in ePTFE electrodes whilst maintaining $C_{2+}$ product selectivity.

## Results

### Local hydrophobicity drives $C_{2+}$ selectivity of ePTFE electrodes

Contrary to ePTFE electrodes, carbon GDL's have the advantage of being conductive (albeit with a much higher through-plane conductivity than the in-plane conductivity)[25,26]. The disadvantage of these

buffer in the alkaline reaction medium to form carbonates and bicarbonates[5]. Second, the accumulation of carbonates and cations close to the cathode catalyst causes the precipitation of carbonate salts that hamper transfer of reactant to the electrocatalytic phase[6]. Third, the hydrophobicity of the carbon gas-diffusion layer (GDL) of the GDE declines (i.e., the carbon becomes more hydrophilic) as current flows through the GDL[7,8], and together with precipitation of hygroscopic carbonate salts, enhances flooding of electrolyte into the GDE pore structure. The flooding of GDE pores with liquid electrolyte blocks gas diffusion pathways for CO₂, which reduces the availability of CO₂ at the electrocatalytic sites and allows the promotion of the HER[9].

To avoid flooding issues during long-term CO₂RR operation, researchers have aimed to increase the hydrophobicity of gas-diffusion layers (Fig. 1a)[10]. One successful approach to increase GDE hydrophobicity is using super-hydrophobic expanded polytetrafluoroethylene (ePTFE) gas-diffusion layers[11–15]. These GDLs consist of a micro-porous layer (MPL) and backing layer (BL). The MPL is made

electrodes, however, is the flooding of the micro-porous layer. Besides promoting unwanted HER at the micro-porous layer, flooding impedes $CO_2$ transport towards the catalyst layer. Whereas, on an ePTFE electrode, dissolved $CO_2$ reacts directly on the interface of dissolution. A fully flooded micro-porous layer (MPL) poses a considerable barrier of around 20–40 μm between the coordinate of dissolution and the reactive surface, as the alkaline electrolyte will start buffering the reactant before it reaches the catalyst[19].

To illustrate the effect of a flooded MPL, we employed a simple reaction-diffusion model at steady-state for both architectures. Solving the model for current densities of −300 mA cm$^{-2}$ along the gas-liquid phase-boundary, we see that the concentration of dissolved $CO_2$ is severely hampered in the flooded carbon MPL case compared to an ePTFE electrode (Fig. 1d, e, see Supplementary Notes for details on the model's parameters).

Comparing the product selectivity of a carbon-based electrode (Sigracet® 38BB) and an ePTFE electrode (Sterlitech® Aspire QL822) over time highlights the limitations of the carbon support which is more susceptible to flooding due to electrowetting of the carbon support. Here we used a three-chamber flow cell with 1 M KOH as both the catholyte and anolyte. The carbon-based GDE was sputtered with a 200 nm Cu layer, and the ePTFE electrode with a 500 nm one. For these two different electrodes we then ran constant current densities experiments from −10 mA cm$^{-2}$ to −300 mA cm$^{-2}$. Here we observed that the ePTFE electrode exhibited a superior selectivity towards hydrocarbon $C_{2+}$ products (ethylene, ethanol, acetate, propane and propanol) across the board (Supplementary Fig. 1). While the copper catalysts on each electrode support are not identical in morphology or surface area, the disparity in selectivities towards the higher value $C_{2+}$ products shows the influence that architecture design has on the performance of a GDE.

## Operando observation of activity distribution on ePTFE electrodes

The dependence of product selectivity on local availability of $CO_2$ then raises questions on the requirements of a catalyst layer. Further, thicker catalyst layers may deplete dissolved $CO_2$ depending on the operating current density or spatial distribution of $CO_2$ under high single-pass conversion conditions[11]. If a thinner and thicker catalyst layer can achieve similar performance, then less material would be beneficial from a cost and resource perspective. The development of stable, thin catalytic layers would then enable a more efficient $CO_2$RR process. However, a drawback of these thin layers is the limited in-plane current collection in the absence of conductive gas-diffusion layers.

To study the effect of this limited in-plane current collection on thin copper films, we deposited two different thicknesses of catalyst on our ePTFE GDLs. Copper thin films deposited by direct-current (DC) magnetron sputtering (see "Methods") show a considerably higher current resistivity compared to bulk metals (e.g. smooth thin-films below 500 nm is up to ~20 times higher in electrical resistance)[27,28]. In the conditions at which these electrodes are operated, i.e. high polarization and high reactant availability, these thin films can then lead to disparity of the current distribution across the GDE. This is directly evidenced by an electrochemical impedance spectra (EIS) analysis which shows the comparatively higher ohmic resistance of 50 nm Cu layer (1.1 Ω) vs. 500 nm Cu layer (0.45 Ω) on ePTFE, as well as vs. the pristine carbon-based support (0.6 Ω) (EIS data in Supplementary Fig. 2, two probes arrangement data in Supplementary Table 3).

Then, using our IR thermography set-up (Fig. 2a, Supplementary Figs. 3–6), we characterized the evolution of temperature distribution during a linear polarization of the cathode. Infrared thermography is a suitable method to study the local activity on thin GDEs, with which one can relate the increase in temperature of the backbone to the local

current density on the catalyst of the electrode[24]. The heat map directly relates to the current distribution profile, hence it offers a powerful proxy to track the current/voltage disparity by cross comparison of GDEs and differing catalyst layers (see the Supplementary Notes). Figure 2b, c shows that the thermographs, especially at higher current densities (<−100 mA·cm$^{-2}$), show a uniform temperature distribution on the backbone of catalyst layers for 500 nm and 1 μm thicknesses. The 50 nm layers, on the other hand, display an activity pattern that is centered around the edges of the electrode for −50 and −200 mA·cm$^{-2}$, where the travel length for electrons from the current collector is much shorter than for the center of the electrode (Fig. 2b, c, right panels). While the surface-wide, averaged temperature increase for both electrodes is similar (ΔT ≈ 4.5 K, see Supplementary Fig. 7), the variance of these temperatures across the surface is considerably more exaggerated for the thinner, less conductive 50 nm Cu layers. This observation implies that, while total activity is similar, as the applied current is the same, the distribution of this activity on the catalyst surface is very divergent.

This irregular electrochemical activity distribution has direct consequences for the observed product distribution. As mentioned before, since the product distribution on copper catalysts is highly dependent on local pH and overpotential, a non-uniform activity pattern is likely to result in an altered product composition[25]. Looking at the selectivity towards $C_1$-products (methane and formate) versus that of higher hydrocarbons, we see that the $C_1$:$C_{2+}$ selectivity ratio of the 50 nm layer results is close to 2: 3 at −200 mA cm$^{-2}$, whereas the thicker 500 nm layer has an approximate 1: 35 ratio (Fig. 2d, Supplementary Table 4).

The trend for the 50 nm catalyst layer, however, is contrary to reports in literature. Upon increasing local overpotential and current density, Cu layers have been observed to shift production to higher hydrocarbons, like ethylene and ethanol[23,29,30]. The observed shift in product distribution can then, at least partially, be explained as a consequence of changing $CO_{2(aq)}$:$CO_{(aq)}$ ratios[31,32]. If we translate the recorded temperatures to a current density value for the observed temperature increases (see Supplementary Fig. 8), local current densities in the thin catalyst layer are then as high as −1 A cm$^{-2}$ at the electrode's perimeter, in contrast to the average applied current density of −200 mA cm$^{-2}$. The elevated local current density then consequently results in greater local depletion of reactant species, lower CO availability and increased overpotential then all play a role in shifting selectivity towards methane and hydrogen[33,34]. In effect, if we use the local current density and temperature values to approximate a concentration for dissolved $CO_2$ using parameters detailed in the model (Equations S9 to S13 in the Supplementary Notes), a distribution ranging from 6 to 10 mM is observed for the 50 nm electrode (Supplementary Fig. 9), where high-activity areas have a clearly lower $CO_{2,aq}$ concentration compared to inactive areas.

## Copper dissolution governs deactivation mechanism of ePTFE electrodes

A potential consequence of the high degree of spatial temperature variances described in Fig. 2 and the preceding section is that the non-uniformity could impact the stability of thin Cu catalyst layers during extended operation and through dynamic electrical loads on the system. Degradation of copper catalysts during $CO_2$RR is a broad concern because changes in the catalyst properties impact the selectivity, activity and stability of electrolysis[35,36]. Several degradation mechanisms have been reported for metallic copper electrodes, like detachment and dissolution[37], Ostwald-ripening[38], reshaping of nano-structures[39] or agglomeration[40]. Restructuring of the copper surfaces has been shown to increase hydrogen production and loss of selectivity towards $CO_2$RR-products[41]. However, it is likely that many of the reported experiments were at conditions (e.g., low current density, H-cell reactors) that facilitated uniform current distributions, such that

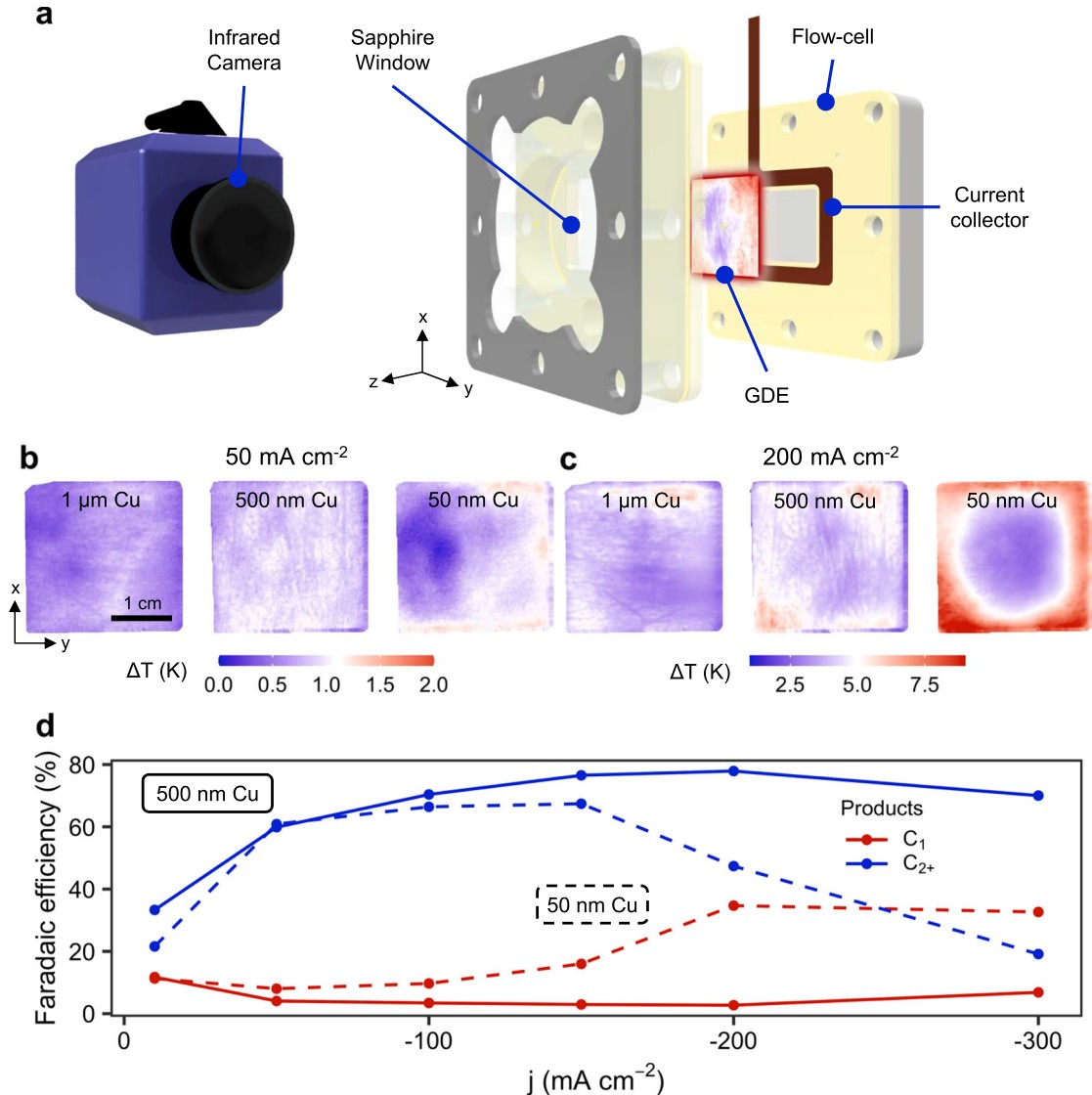

**Fig. 2 | Current distribution in an ePTFE electrode strongly depends on catalyst thickness. a** Infrared thermography setup for an electrochemical flow cell. **b** Space-dependent temperature increase on an ePTFE/1000 nm Cu electrode (left), an ePTFE/500 nm Cu electrode (center), and an ePTFE/50 nm Cu electrode (right) at −50 mA cm⁻². **c** Space-dependent temperature increase on the same electrodes at −200 mA cm⁻². **d** Impact of catalyst-layer thickness on the product selectivity of ePTFE electrodes at increasing current densities.

the effects of significant variations in current density across the electrode were not considered. Further, many prior studies used planar-type electrodes that do not represent the complexity of the three-dimensional, multi-layered porous GDE structures like the ePTFE electrodes presented in this study.

We hypothesized that copper is exposed to spatially varying effects for more complex and larger surfaces at abrupt interfaces, like the ePTFE electrodes presented herein. Under the absence of polarization or reduced electrochemical potential, copper forms hydroxide species in basic conditions, as is the case for the electrodes described in this work[42]. Ultimately, this means degradation of the catalyst layer is accelerated under alkaline $CO_2RR$ conditions. In effect, even the thicker 500 nm Cu layers in this work presented loss of selectivity for $C_{2+}$-products and increased $H_2$-production rates in a short period (Supplementary Fig. 10).

To scope and assess the degradation mechanisms for the thin 50 nm Cu layers, we subjected our electrodes to a slow polarization increase. Combined with the IR thermography, this allowed us to determine an operando activity distribution (Fig. 3a and b). For low polarizations, like the −25 and −50 mA cm⁻² cases, the temperature

increase compared to open-circuit voltage is evenly distributed over the surface, which indicates a homogeneous current-density distribution. Upon increasing this polarization to higher values, the activity quickly accumulates to those areas closest to the current collector, where the path of electric resistance is lowest. This effect starts manifesting beyond −50 mA cm⁻². Figure 3c shows that the area around the edges of the electrode have an increased temperature gradient (red), whereas the center part of the electrode stagnates (blue).

The lack of polarization at the center of the electrode presents an opportunity for copper to dissolve in the locally high alkaline environment. All sites subjected to a polarization of less than −0.5 V vs. SHE are prone to the formation of $Cu(OH)_2$ species, which easily transforms to $CuO$[42,43]. In effect, optical examination of the catalyst layers before and after electrolysis shows an increased degradation for thinner films (see Supplementary Figs. 11 and 12). XPS-analysis of both 500 nm and 50 nm samples, before and after polarization as aforementioned are displayed in Fig. 3d. Pristine samples of both thicknesses show the characteristic Cu $2p_{3/2}$ peak at ~933 eV and a Cu $2p_{1/2}$ one at 952 eV[44]. Meanwhile, the pristine 500 nm sample shows a distinctive shoulder peak at 935 eV. After polarization, on the other hand, both samples

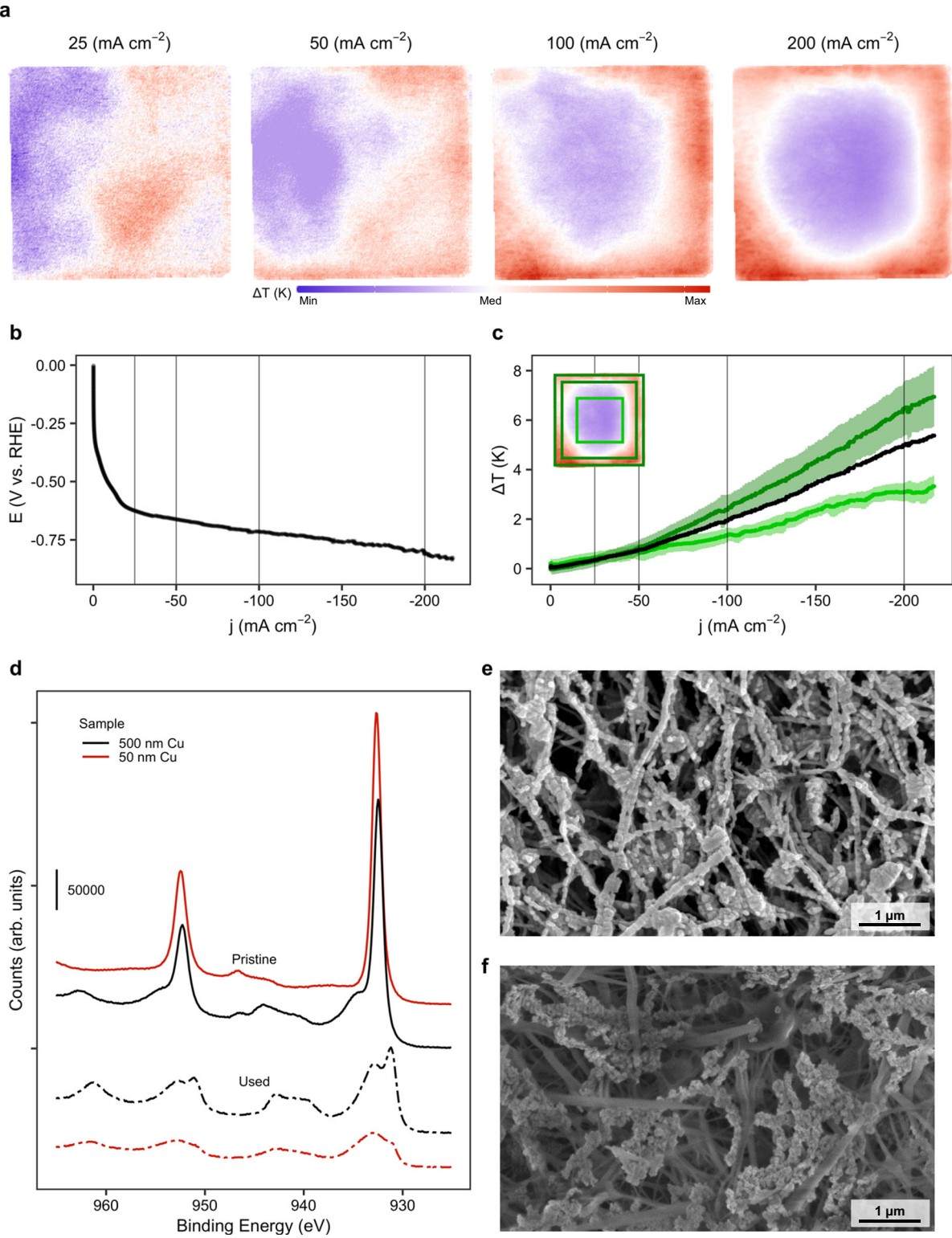

**Fig. 3 | Degradation of ePTFE/Cu electrodes is highly dependent on local polarization. a** Activity distribution in an ePTFE/50 nm Cu electrode at increasing current densities. **b** Polarization curve for the electrode in (**a**). **c** Observed average temperature increase and its deviation for the high and low activity zones in (**a**). Electrode average temperature in black. Insert depicts the thermograph at −200 mA cm⁻² and the analyzed areas. **d** XPS spectrograms of pristine and used ePTFE/500 nm and ePTFE/50 nm samples. **e** SEM image of a pristine ePTFE/50 nm sample. **f** SEM image of a used ePTFE/50 nm sample after 2 h of operation.

have a reduced Cu $2p_{3/2}$ peak and two shake-up peaks developed at 942 and 963 eV, similar to $Cu(OH)_2$ scans[45]. The lower intensity of copper-specific peaks and the presence of the soluble hydroxide species suggest that copper mainly detaches from the surface by dissolution in the basic electrolyte. This is further confirmed by Cu LMM scans, which

show the presence of a metallic underlayer for pristine samples[46]. This metallic underlayer is preserved in the 500 nm Cu sample, but missing in the 50 nm sample (see Supplementary Fig. 13).

To strengthen this analysis, we proceeded to study both samples under SEM and AFM. Figure 3e, f depict the surface of the 50 nm

electrode before and after electrolysis, respectively. The conformal coating of the expanded PTFE fibers by copper in Fig. 3e matches earlier reports in literature[18]. An analysis of the used sample showed little remnants of copper on the surface close to the center of the electrode, which could be fully oxidized. The increased difficulty in acquiring these later images also confirms the reduction in electrical conductivity of used 50 nm samples. While 500 nm samples show a minor degradation after running (see Supplementary Figs. 14–16), there seems to be little detachment of catalyst from the electrode (e.g. ~40 nm or 17% copper film thinning around a fiber calculated from measured average of 234 to 195 nm, before and after $CO_2RR$). This contrast in detachment is confirmed by AFM scans we performed on 50 nm Cu ePTFE electrodes and 200 nm Cu carbon electrodes (see Supplementary Figs. 17, 18).

On the other hand, selectivity figures for high current densities show a loss of selectivity and progressive deactivation of electrodes. The excessive equivalent current densities shown in Supplementary Fig. 8 point to cathodic corrosion and restructuring as a primary deactivation mechanism at areas close to the current collector[47,48]. An increased local potential on copper nanoparticles is known to result in accelerated restructuring of the surface, resulting in stabilization of certain intermediates and increase in partial current densities towards hydrogen[37,49].

All in all, it appears that the physical stability of copper electrodes is highly dependent on its conductivity and exposure to a certain potential value to ensure that the catalyst would remain in a fully reduced, metallic state. Within 30 min, the 50 nm sample would develop areas that are or poorly connected (or even disconnected) from the current collector leading to chemical oxidation and dissolution of the copper species. Conversely, the areas that remain electrically connected near the perimeter of the electrode would experience accelerated deterioration due to the increased share of the local current density (Supplementary Fig. 19). This ultimately means that any solution that ambitions to stabilize copper electrocatalysts for $CO_2RR$ must take the equalization of the current distribution across the surface into account.

## A non-invasive current collector improves stability of ePTFE electrodes

Reports of using of ePTFE electrodes for $CO_2RR$ generally involve a small catalytic surface area, in the order of 1–2 cm$^2$ [11,18]. On top of this, these catalyst layers are usually in the order of a couple of microns, instead of the much thinner hundreds of nanometers traditionally used in the deposition on carbon-based GDEs[9,50–53]. As we have shown above, the stability of the copper in the reaction environment not only depends on previously mentioned degradation mechanisms, but also on its corrosion if the current collection is insufficient. Any solution to the stability of copper catalysts on super-hydrophobic substrates goes, then, through solving the current collection issue.

Previously developed electrodes using these super-hydrophobic backbones have overcome current collection issues simply by applying a rather thick sandwiching layer consisting of carbon nanoparticles (C-NPs) and a graphite coating on top[11]. The use of carbon, however, implies the risk of promoting HER if sufficient potential is applied on the cathode. This solution seemed to stabilize electrode performance for operational times upwards of 100 h, but reported results are at rather low current densities and using highly conductive catholytes. While this solution was sufficient to achieve the metrics reported, it seems unfeasible to apply the same solution at a larger scale for high current densities and less concentrated electrolytes. Under these conditions, the applied potential is likely to surpass the −0.8 V vs. RHE required for HER promotion on the graphite and C-NPs at high rates[9,54,55]. On top of this, an added current collection layer between the cathode and anode is likely to result in longer ionic

pathways, increasing ohmic resistances in the system and, in the case of $CO_2RR$, accumulating and precipitating (bi)carbonate salts.

In designing a remedy for the copper dissolution, we turned our attention to non-invasive solutions. This means a design that is intrinsically non-invasive and has minimal effect on the overall product distribution of our catalyst. As we highlighted before, thin copper catalyst layers have a limited conductivity when compared to the bulk metal. The usage of non-invasive bus-electrodes (or busbars) is widespread in the manufacture of photovoltaic semiconductors. Here, a thin busbar of (most commonly) silver collects current generated by the semi-conductor upon exposure to sunlight[56,57]. By segmenting an area into smaller current collection channels, the ohmic resistance experienced by current traveling in-plane is greatly reduced. To illustrate the comparative conductance of different copper film thicknesses, we measured the ohmic drop of a 1 μm Cu film using a 2-electrode probe showing a 9-fold reduced resistivity for the 1 μm vs 50 nm thin films (see Supplementary Table 3).

Making use of deposition masks, we sputtered 1 μm thick copper busbars on a 50 nm Cu/ePTFE electrode to fabricate non-invasive current collectors (NICCs) for our electrode architecture (see Fig. 4a, Supplementary Figs. 22 to 24, and Supplementary Table 5). These busbars made electrical contact with the front-sided copper tape in the flow cell (while the copper tape is isolated from the catholyte, so it does not contribute to the catalyst area), functioning in practice as 'highways' for electrons to travel through before spreading out over the catalyst surface. The intention, in terms of system design, of deploying this solution is twofold: first, the low profile of these current collectors avoids any leakage of electrolyte between the gasket and the GDE and presents a facile, scalable design; and second, the material acting as collector is the same deployed as catalyst, which should avoid excessive promotion of unwanted side-reactions, like HER. The goal of these NICCs is not to prevent electrochemical reactions on the 1 μm copper busbars, but to minimize their effect on product mixes while improving overall electrode current density distribution.

To test the efficacy of such an approach, we decided to compare to the previously benchmarked 50 and 500 nm catalyst interfaces. In effect, when comparing the latter two with a Cu/NICC sample, the thermal signature of the NICC enabled electrode is much more even than that of a bare 50 nm sample, much like that of the 500 nm one, as shown in Fig. 4b. This means, in short, that our design results in an improved current collection when compared to traditionally deposited thin-films on ePTFE GDLs.

The current collection and distribution of activity over the catalytic area translates itself into an observable improvement of selectivity and stability for thin catalyst-layers. Comparing all three designs (500 nm, 50 nm and 50 nm/NICC on an ePTFE electrode) at constant potential during extended periods (4 hrs. max.), resulted in the ethylene selectivity displayed in Fig. 4c. Even though the onset selectivity of the NICC design is lower compared to the former 500 nm layer, a steep drop was observed for the films using the conventional method. To note, after 4-h of electrolysis, NICC electrode can preserve 30% FE for ethylene whereas the conventional electrodes lose their selectivity, dropping below 10% FE.

While these reported stability durations are lower than state-of-the-art reports[6,18], the mass loadings of copper in this work are orders of magnitude lower (0.112 mg$_{Cu}$ cm$^{-2}$ for a 50 nm Cu/NICC electrode versus ~ 2 mg$_{Cu}$ cm$^{-2}$). This observation is supported by 7-h constant potential measurements comparing (i) a 1 μm Cu electrode, (ii) a 200 nm Cu/NICC electrodes, and (iii) a 500 nm Cu sample (Supplementary Fig. 25) indicating higher mass loadings are an enabling factor in stability studies. Our results however still demonstrate the added stability towards $C_2H_4$ of the 50 nm/NICC design versus the 500 nm and 50 nm counterparts (see Supplementary Table 6), though copper stability that plagues the $CO_2$ and CO reduction fields remains.

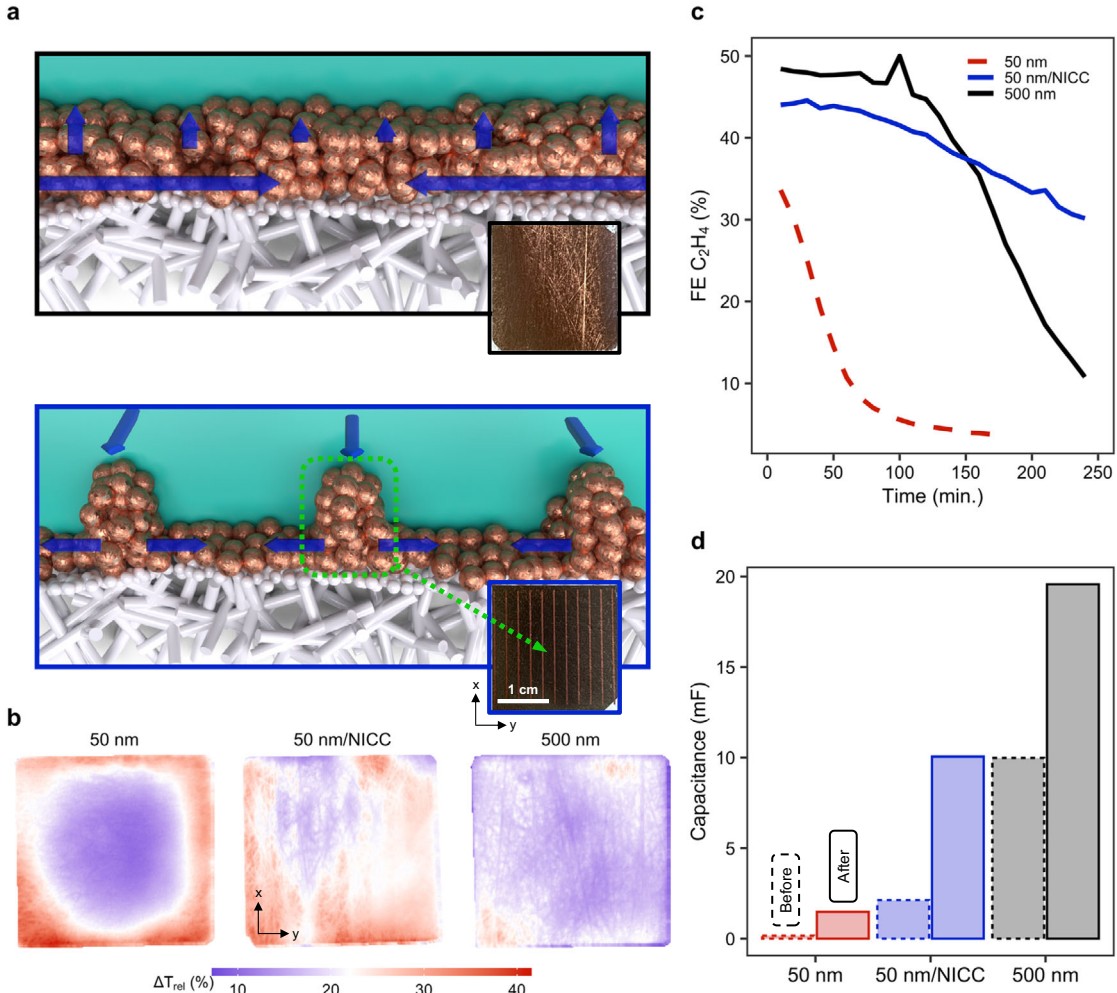

**Fig. 4 | A non-invasive current collector extends stability and selectivity of ePTFE/Cu electrodes. a** Sketches of a traditional ePTFE/Cu (top) and ePTFE/NICC/Cu electrode (bottom). Current collection lines depicted in blue. **b** Activity distribution for an ePTFE/50 nm, an ePTFE/50 nm/NICC and an ePTFE/500 nm electrode at −200 mA cm$^{-2}$. **c** Observed selectivity towards ethylene for the three designs at a constant potential of ~ −0.55 V vs. RHE. **d** Change in capacitance for the three designs after 4h-long operation at constant potential of ~ −0.55 V vs. RHE.

As the busbars themselves are made of copper, it is also interesting to examine if the electrochemical activity for the 50 nm/NICC case can be attributed to the 1 μm Cu busbars, as compared to the 50 nm Cu layer over the entire electrode. For instance, we model that sufficient dissolved $CO_2$ is expected to be present at 1 μm depths even at higher current densities (Fig. 1d). To understand this influence, we can compare the ethylene Faradaic efficiency for the 50 nm Cu case, the 50 nm/NICC Cu case, and an electrode only containing the 1 μm Cu busbars (e.g. ePTFE + NICC). As shown in Supplementary Table 5 and Supplementary Fig. 20, the 1 μm Cu busbars show a quick decline in ethylene selectivity with increasing current density, while the combined 50 nm/NICC Cu case maintains ethylene selectivity up to −200 mA cm$^{-2}$. We can then conclude that the greater performance of the 50 nm/NICC system over the 50 nm sample alone is not due to the additional mass loading of the 1 μm Cu busbars.

A way of monitoring the evolution of surface roughness during electrolysis is the measuring of double-layer capacitance in the used GDEs[58]. To do so, we performed cyclic voltammetry before and after long electrolysis runs (~4 h) at different scan rates between −0.3 and 0.3 V vs. Ag/AgCl. The capacitance analysis of the NICC design shows a noticeable increase against its 50 nm counterpart (Fig. 4d). The thicker, 500 nm electrode, however, still displays a bigger capacitance overall. The change in capacitance is relatively smaller for the 50 nm/NICC and 500 nm designs than for the 50 nm one, indicating a

dramatic increase in surface roughness. Compared to the 50 nm/NICC design, the 500 nm layer displays a lower capacitance change (around two-fold, 10 mF before and 19.5 mF after). This indicates that, while corrosion of the electrode can influence selectivity directly, it is not the main driver of selectivity changes, as the 50 nm/NICC shows an improved selectivity towards ethylene over time. While the sharp disparity between before and after measurements can also be due to the presence of oxide species in the fresh samples tested before electrolysis, the noticeable increase for the 50 nm/NICC sample suggests a considerable larger area of the electrode is electrically connected. The improved current density distribution and more equal overpotential distribution, then, appears to also influence degradation of these electrodes directly.

## Discussion

We will now briefly comment on the NICC approach more broadly for membrane electrode assembly systems, and the potential benefits of replacing the copper busbar structure with silver.

It is worthwhile to discuss the potential for the NICC in membrane-electrode assembly systems which still predominantly utilize carbon-based GDE's. Firstly, while carbon-based GDE's can maintain long term $CO_2$RR performance in MEA systems due to the lack of catholyte, operational wetting and flooding of the MPL layer is still occurring as evidenced by salt precipitation on the back of the GDE.

Further, modeling studies in the fuel cell domain have shown the preference of previously wetted MPLs and GDLs to remain wetted, as removing water from nanopores is challenging[59]. These wetted areas then likely act as an impediment for $CO_2$ gas flow, even if not impeding it entirely. Additionally, flooded regions act as isolated liquid volumes susceptible to anion and cation concentration reaching precipitation levels, further hampering mass transport. Such periodic water volumes, salt deposits and condensate release within carbon GDL's are shown by X-ray experiments, and may be prevented using the NICC with an ePTFE backbone[60–62]. Overall, we hypothesize that the use of carbon-based GDE's in MEA systems may be one reason why the CO selectivity of many silver-based zero-gap MEA systems is predominantly limited around −200 to −300 mA cm$^{-2}$ while the same GDE and catalyst in flowing catholyte systems reaches much higher $CO_2$ reduction current densities before HER becomes dominant[18,63]. Efforts to utilize ePTFE electrodes and the NICC in an MEA systems can be of interest for future studies to overcome these limitations.

An interesting replacement for copper busbars in future studies would be the use of silver busbars as used in photovoltaics. Silver provides the best conductivity of any pure metal, with 5% greater conductivity than copper, which would reduce voltage losses along the busbar. Any by-product carbon monoxide formed during $CO_2$ reduction on the silver would also likely be utilized as a reactant on the copper layers. Finally, silver is a more stable electrode material than copper, likely providing greater longevity. For these reasons future work should consider silver as a busbar material.

The demonstrations in this work were on a 2.25 cm by 2.25 cm (5 cm$^2$) electrode. To examine the prospect of using the ePTFE and copper busbar approach for larger electrodes, we assessed the busbar dimensions required for a 20 cm by 20 cm (400 cm$^2$) electrode to maintain an identical voltage drop from the exterior current collector to the center of the electrode. These calculations are shown in the Supplementary Notes and in Supplementary Fig. 21 which indicate the NICC approach is feasible for larger electrodes with a busbar spacing of 20 mm, a busbar height of 100 μm, and a busbar width of 3 mm. These busbar dimensions could further be reduced using cross-hatched patterns or reduced busbar spacings.

In conclusion, we have shown spatial and temporal implications of current collection on the performance of thin catalyst layers for $CO_2RR$. Particularly in the case of non-conducting ePTFE gas-diffusion layers, targeted efforts are needed to provide current collection pathways across the entire catalyst layer, due to the conductivity limitation of <500 nm thick catalyst layers. By illustrating the spatial degradation of a 50 nm copper catalyst layer using infrared thermography, we built a better understanding of the effects of catalyst migration on spatial temperatures, current densities, and reactant availability. Further, by adding a non-invasive current collector to the 50 nm Cu-catalyst layer, we showed that the electrode's performance can approach and exceed 50 nm and 500 nm conventional films. This opens the pathway to further advancement of thin-layer catalysts on ultra-hydrophobic GDEs for $CO_2$-electrolysis.

Especially for ePTFE electrodes, which hold a potentially important role in advancing $CO_2RR$, current collection has long been an understudied subject, mainly because of unobserved limitations in lab-scale electrolyzers. Without addressing issues surrounding current collection on ePTFE supports, these gas-diffusion layers are unlikely to be industrially applicable. Beyond this work, greater considerations about current collection in catholyte flow field channels, and how to connect ePTFE electrodes in stacked configurations are critical. This work begins these discussions, hopefully opening the door to future work.

## Methods

### Cathode fabrication
Copper metallic layers were deposited directly on the ePTFE (Sterlitech® Aspire QL822, 0.45 μm) or carbon-based GDLs (Sigracet® 38BB) by DC magnetron sputtering at a pressure of 3 μbar. By regulating the power applied and the time of exposure, we were able to regulate the thickness of the catalytic layer. After deposition, the electrodes were stored in a glovebox (<0.1 ppm $O_2$ and <0.1 ppm $H_2O$) and only taken out before assembly of the electrolyzer cell. NICCs were sputtered using a slit-mask with 0.3 mm wide, 1 μm thick Cu slits along the entire width of the pre-sputtered 50 nm ePTFE GDE.

### Flow-cell design
We employed proprietary flow-cell design that is based around a commercially available titanium anode block (Dioxide Materials®, 5 cm$^2$ Titanium Anode Block). Around this anode block, we designed and printed a catholyte flow-chamber with a reference electrode port (∅ = 1.75 mm) and a similar gas-chamber, with matching gaskets as sealants (see Figs. S5 and S6). Current collectors for ePTFE cathodes were copper-tape based, precision-plotted using a Cricut® Maker 3. The assembly was completed with a gas/electrolyte end-block with an incorporated infrared window, and a metallic pressing plate.

Experiments were performed using untreated nickel (Ni) foam as the anode, and Sustainion® X37-50 anionic exchange membranes (AEMs). Electrolyte was flown pumping the electrolyte and sucking the anolyte, as to generate a pressure delta across the membrane. This avoided expansion of the membrane in the catholyte compartment and accumulation of $O_2$-bubbles in the anolyte compartment.

### IR-thermography setup
Thermal images of the backbone of the electrode were acquired using a FLIR SC7650 camera system, using the windowed electrolyzer. The camera was equipped with a fixed focal length of 25 mm and acquired images at a framerate of 1 fps with a total resolution of 640 by 512 pixels, operated by commercial ALTAIR® software. The total scanned area of the electrode was 5 cm$^2$ (±2.25 by 2.25 cm square), with an Edmund Optics® uncoated sapphire window (δ = 1 mm, ∅ = 32 mm). Temperature values acquired with the camera were corrected by the compound transmittance of the sapphire window and the emissivity of the polypropylene backing of the ePTFE electrodes. Measurements were performed under a light-shielding blanket to avoid contamination of signals with external reflections on the camera lens.

### Electrochemical setup
Electrochemical routines were applied using either a Princeton Applied Research® Parstat 4000 (±48 V, 20 A) or MC-1000 (±12 V, 2 A) in a three-electrode configuration. Gas-flow to the electrolyzer was controlled by a Bronkhorst® mass-flow controller. Electrolyte was pumped using two separate peristaltic pumps and two sets of pressure dampeners, to alleviate the cyclic pressure spikes of the peristaltic motion. All three fluid channels were regulated through a back-pressure regulator (BPR), which controlled pressure for each stream independently, and measured live mass flowrate for the gaseous product stream. This stream was circulated through a liquid trap and a proprietary liquid detector to avoid liquid injections to the in-line gas-chromatographer (GC, Global Analyser Solutions® CompactGC 4.0). Before reaching this GC, a valve regulated pressure spikes to avoid disruption of the backpressure after each injection. Liquid product samples were collected periodically in duplicate from the catholyte beaker using a needle and were stored in air-tight vials at 5 °C to avoid evaporation of volatile species. These samples were then analyzed in batch using an Agilent Technologies® 1260 Infinity II HPLC.

Polarization curves were applied by a galvanodynamic swing from 0 to −300 mA cm$^{-2}$ at a rate of −1 mA cm$^{-2}$ s$^{-1}$. Product distribution studies were acquired either at increasing fixed current densities or at fixed working electrode potential (without i·R feedback correction). Capacitance measurements were performed on all three different architectures by cycling the electrodes from −0.3 to +0.3 V vs. Ag/AgCl, collecting the average charging current over a non-Nernstian region

for the anodic swing at increasing scan-rates (5, 10, 15, 20 and 25 mV s$^{-1}$). The value for capacitance was extracted as the slope of the linearization of charging current over scan-rate.

## Characterization

Pristine and post-operation samples of both carbon and ePTFE GDE's were stored in a glovebox before characterization. Scanning electron microscopy (SEM) was performed on a JOEL JSM-7001F and FEI Nova NanoSEM 450 instruments. To improve image quality, the surfaces of GDE were cleaned using an Evactron 25 De-Contaminator RF Plasma Cleaning System, and the ePTFE GDEs were coated with a Pt film (~5 nm) using a Quorum Q150T Metal Coater. One more plasma cleaning was conducted after the coating process. In addition to this, a fresh cross-section of GDE was obtained by breaking the sample in liquid nitrogen before the general sample preparation.

XPS spectra were collected using a Thermo Scientific® K-alpha spectrometer using an Al Kα monochromator. For all measurements (F1s, C1s, Cu LLM, Cu2p and valence-band scans) the spot size was 400 μm, the pass energy 50 eV and the step-size 0.10 eV, while the base pressure of the analysis chamber was $2 \cdot 10^{-9}$ mbar. Resulting scans were averaged after 10 measurements (valence bands required the averaging of 50 scans).

## Data availability

The data that support the findings of this study are available from the corresponding author upon reasonable request.

## Code availability

The code and resulting data files that support the findings of this study are available from the corresponding author upon reasonable request.

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

## Acknowledgements

H-P.I.v.M. and T.B. would like to acknowledge Joost Middelkoop for the assistance in designing, plotting and printing parts for the windowed electrolyzer, Herman Schreuders for suggesting an approach similar to a 'busbar' electrode and help with sputtering catalysts, and Reinier den Oudsten-Grijzen for assistance in designing, milling and commissioning parts for the electrolyzer and the sputtering masks. H-P.I.v.M. and T.B. acknowledge the financing provided for this project in the context of the e-Refinery Institute by Shell Global Solutions International B.V. and the Top Consortia for Knowledge and Innovation (TKI's) of the Dutch Ministry of Economic Affairs. ML acknowledge the financial support from Australian Research Council (DE230100637). Y.W. and T.R. acknowledge the facilities, and the scientific and technical assistance, of the Australian Microscopy & Microanalysis Research Facility at the Center for Microscopy and Microanalysis, The University of Queensland.

## Author contributions

Conceptualization: H.P.I.v.M. Methodology: H.P.I.v.M., T.B. Investigation: H.P.I.v.M., M.A., S.S., M.S., E.I. Characterization: H.P.I.v.M., M.L., Y.W., S.K.P., D.R. Visualization: H.P.I.v.M., M.L., S.S., J.B., T.B. Funding acquisition: T.B. Project administration: H.P.I.v.M. Supervision: T.B. Writing—original draft: H.P.I.v.M. Writing—review & editing: H.P.I.v.M., T.B., M.L., J.B., T.E.R., M.A., E.I.

## Competing interests

H.-P.I.v.M. and T.B. have filed a patent application (no. 2028898) concerning the method and configuration for observing catalytic activity via infrared radiation. The remaining authors declare no competing interests.
