## [Peer Review File · Nature Communications]

Non-invasive current collectors for improved current-density distribution during CO₂ electrolysis on super-hydrophobic electrodesREVIEWER COMMENTS

Reviewer #2 (Remarks to the Author):

Summary:

In this manuscript, the authors use non-interfering current collectors, much like the bus-bars of photovoltaic cells, to resolve the limited in-plane conductivity of thin catalyst layers on PTFE gas diffusion layer substrates. The authors use operando experiments to understand the poor current distribution of thin catalyst layers and compare against thick catalyst layers and their current collector system. I appreciate the creative strategy and inspiration from a different technology, but would like to see more depth in the dataset as described in comments below.

Major Comments:

- ePTFE gas diffusion layers are commonly used in lab-scale experiments in flow-cell electrolyzers where there is a flowing catholyte. However, recent shifts to membrane-electrode assembly (MEA) electrolyzers (for their lower cell voltage, improved stability, and more relevance to scale-up) has seen a return of high-PTFE content carbon-based GDLs. In this case, the entire electrode is conductive and there is lower risk of flooding without the catholyte layer. Is the NICC limited to flow cell electrolyzer, or does it offer any benefits to MEAs? Please comment on this topic.
- The paper describes how the in-plane resistance increases as electrode area increases, but all of the data in the paper seems to be on 5 cm² electrodes. Since improving current collection at larger sizes for scalability is critical to the motivation of this electrode design, I would strongly encourage the authors to collect data using larger electrodes and show with evidence the drop-off in performance vs size as a result of poor current collection, and then also show with data the improvement using their design at larger electrode areas.
- Looking at Fig. 4, since the bus bar is also copper, is it not also catalytically active? If so, I suggest re-evaluating the use of "non-interfering" since it would be involved in the reaction. Furthermore, since the copper on the bus-bar is significantly further (2 orders of magnitude) from the GDL than the Cu in between bus-bars, I request some explanation as to how the local chemical micro-environment differs between these catalytic sites of the electrode and how that is connected to the overall product distribution of the electrode. Sensitivity analysis experiments changing the height of the bus bar may help to assess the impact.

Minor Comments:

- In the title, NICC = "non-invasive current collector", but in the abstract, NICC = "non-interfering current collector". It would be good to stay consistent with the acronyms that you define.

Reviewer #3 (Remarks to the Author):

In this work, Hugo-Pieter Iglesias van Montfort and colleagues investigate current collection on ePTFE GDLs for CO₂ electrolysis in 1 M KOH. They first compare carbon paper and ePTFE electrodes on the basis of product selectivity where they show that flooded carbon MPLs make less multicarbon products due to a reduced conversion of the low solubility CO intermediate. The authors then use an IR thermography setup to map the reaction rate on ePTFE electrodes with 50 and 500 nm thick catalyst layers. They show that the thinner catalyst layer suffers from uneven catalyst utilization with areas near the edges receiving the most activity. They perform material characterization (XPS, SEM, and AFM) to suggest that copper dissolution at less reducing potentials is the primary cause of instability. To improve conductivity, they sputter copper busbars across the face of their electrode which improves their stability when tested over a four-hour span.

This is an interesting idea which merits consideration for publication if the following major concerns can be properly addressed:

1) Catalyst detachment: The authors devote a section of their paper to demonstrate that the centre of the 50 nm Cu ePTFE electrode experiences catalyst detachment because it is not under a sufficient reducing potential. If these catalyst sites are not being used for the reaction, the authors should explain how their detachment can be the cause of the electrode instability. The authors focus on the catalyst material which is not being used for the reaction. Could the active catalyst material (which is at a sufficiently reducing potential to protect the copper) be experiencing accelerated degradation since it is being overworked? Salt precipitation is another potential failure mechanism and it would be reasonable to assume that high local reaction rates drive up the pH and accelerate salt precipitation. This manuscript could benefit from further analysis or exploration to explain the cause of instability amongst the active catalyst sites.

2) Stability benchmarking: The authors should provide a better explanation as to why their stability (only 4 hours) is so much less than other ePTFE electrodes demonstrations (>100 h). They hint that it is due to the lower electrolyte conductivity or higher current density (I suspect it is predominantly the current density at play) but I would suggest the authors run some control experiments demonstrating how stable their electrode could be if run in similar conditions to these other works. This could also assist in the discussion on my first concern above as they identify what is causing their electrodes to fail.

3) Non-intrusive nature of NICC: The authors claim carbon support layers used in previous electrode designs could evolve hydrogen at a reasonable operating voltage, so they instead they chose a copper busbar material which they claim is non-interfering. However, copper can also evolve hydrogen in the absence of CO₂ and the busbars are very thick (1000 nm) compared to the catalyst layers (50 or 500 nm). The current work does not persuade me that these copper busbars are non-intrusive. Have the authors done any modelling to predict what the CO₂ concentration is on these busbars deep in the electrolyte? Can they demonstrate experimentally how much (or how little) reaction current is taking place on the busbars? How would the voltage distribution and stability be impacted if another material was used (silver would be an interesting comparison since it is a poor catalyst for hydrogen evolution)?

4) 1000 nm catalyst layer control experiment: The manuscript could benefit from a stability, activity distribution, and selectivity assessment of an ePTFE electrode with a 1000 nm catalyst layer (the same thickness of the busbars, but applied uniformly across the electrode surface). The authors should share the polarization data for the 1000 nm, 50 nm, 500 nm, and 50 nm/NICC electrodes.

I also have the following minor concerns:

1) Selectivity: The hydrogen selectivity for all experiments should be included (at least in the SI tables). Also, the product-specific breakdown of selectivity should be provided for the carbon paper electrode (Figure S1) in tabular form as it is for the other experiment sets.

2) Flooding of carbon electrodes: The authors should show the stability of ePTFE and carbon electrodes. They mention flooded carbon GDLs immediately in the first section but the paper would be stronger if they showed how fast flooding sets in.

3) Activity distribution of the 50 nm/NICC: The authors should indicate, graphically or in the caption, the orientation of the NICC (e.g., horizontal or vertical) as shown for Figure 4B.

4) PTFE vs. ePTFE: In the introduction, the authors should double check when PTFE and ePTFE are used. Here are some examples which I think should be referring to ePTFE: "...non-conducting PTFE

GDE's maintain hydrophobicity during sustained...", "...electrolyzers using PTFE GDE's have then shown the ability...", and "...CO2 diffusion properties of PTFE GDE's...". Perhaps the authors should add a sentence clearly distinguishing PTFE and ePTFE.

5) Figure 1: I would suggest adjusting the colour scheme of Figure 1A to be more friendly towards colour-blind readers.

6) Typos: there are some minor typos in the manuscript (e.g., "...as is the case for the carbon electrode, the maximum local monoxide concentration...").

Reviewer #4 (Remarks to the Author):

The manuscript describes the fabrication of novel non invasive current collectors for improved current density distribution during CO2 electrolysis on super-hydrophobic electrodes.

The development of robust electrodes for CO2 electrochemical reduction is an important topic in the field of CCU as they can be one of the limiting parameters for commercialization of CO2 electrolyzers. The manuscript is well written, contains a significant amount of experimental observations, but in some cases the discussion is based in assumptions and not experimental evidences. Some examples below:

- The operando IR tomography is a valuable technique for evaluating the current distributions, however, it does not provide solid evidences on the factors that lead to that distribution over the electrode surface.
- The interpretation of the differences in activity between the carbon GDL and the ePTFE is very simplistic. As shown in figure S15 and S16, the morphology of the obtained Cu films is very different and by itself can lead to different activities and product distributions.
- As the authors also analyze, there are significant morphological modifications on the electrode surface during reaction (see the relative difference in the capacitance of the electrodes in figure 4, for example). The rate of the surface corrosion can also be the difference between the electrodes with different Cu thicknesses.

The NICC is definitively an interesting approach for the development of gas diffusion electrodes for CO2 but that is in an engineering solution in itself.

Reply to reviewers' comments

Non-invasive current collectors for improved current-density distribution during CO₂ electrolysis on super-hydrophobic electrodes

Reviewer #2 (Remarks to the Author):

“ePTFE gas diffusion layers are commonly used in lab-scale experiments in flow-cell electrolyzers where there is a flowing catholyte. However, recent shifts to membrane-electrode assembly (MEA) electrolyzers (for their lower cell voltage, improved stability, and more relevance to scale-up) has seen a return of high-PTFE content carbon-based GDLs. In this case, the entire electrode is conductive and there is lower risk of flooding without the catholyte layer. Is the NICC limited to flow cell electrolyzer, or does it offer any benefits to MEAs? Please comment on this topic.”

We acknowledge that MEA electrolyzer designs are widely seen as the most [promising configurations for large-scale CO₂ electrolysis (CO₂RR), seeing as the lower cell voltage and stability against flooding are considerably better than flow-electrolyzers. We also agree that high PTFE-content carbon micro-porous layers will likely remain the default GDE in MEA systems. However, we see potential opportunities for the benefits we observed with NICC in our flow-cell electrolyser experiments to be helpful in MEA systems because current distribution and current distribution will also be challenges in large area MEA cells. Therefore, acknowledging this comment, we added the following discussion to the manuscript to provide some perspective for the reader on the future use of NICC in MEA electrolyzers.

Added to pg. 15:

It is also worthwhile to discuss the potential for the NICC in membrane-electrode assembly systems which still predominantly utilize carbon-based GDE's. Firstly, while carbon-based GDE's can maintain long term CO₂RR performance in MEA systems due to the lack of catholyte, operational wetting and flooding of the MPL layer is still occurring as evidenced by salt precipitation on the back of the GDE. Further, modelling studies in the fuel cell domain have shown the preference of previously wetted MPLs and GDLs to remain wetted, as removing water from nanopores is challenging. (A. Weber, Journal of Power Sources 195 (2010) 5292–5304). These wetted areas then likely act as an impediment for CO₂ gas flow, even if not impeding it entirely. Additionally, flooded regions act as isolated liquid volumes susceptible to anion and

cation concentration reaching precipitation levels, further hampering mass transport. Such periodic water volumes, salt deposits and condensate release within carbon GDL's are shown by X-ray experiments, and may be prevented using the NICC with an ePTFE backbone. (Disch, J. et al. Nat. Commun. 13, 6099 (2022), Garg, S. et al., Energy Environ. Sci., 16, 1631 (2023), Moss et al., Joule, 7, 350-365 (2023)) Overall, we hypothesize that the use of carbon-based GDE's in MEA systems may be one reason why the CO selectivity of many silver-based MEA systems is predominantly limited to 200-300 mA cm⁻² while the same GDE and catalyst in flowing catholyte systems reaches much higher CO₂ reduction current densities before HER becomes dominant. (García de Arquer, F. P. et al., Science 367, 661–666 (2020)), (Endrődi, B. et al., Nat. Energy 6, 439–448 (2021)) Efforts to utilize ePTFE electrodes and the NICC in MEA systems can be of interest for future studies to overcome these limitations.

“The paper describes how the in-plane resistance increases as electrode area increases, but all the data in the paper seems to be on 5 cm² electrodes. Since improving current collection at larger sizes for scalability is critical to the motivation of this electrode design, I would strongly encourage the authors to collect data using larger electrodes and show with evidence the drop-off in performance vs size because of poor current collection, and then also show with data the improvement using their design at larger electrode areas.”

We agree with Reviewer #2 that more experimental data for electrolyzers with large-area electrodes will be critical to understanding CO₂RR electrolyzer performance issues and scaling effects in industrial-scale applications. However, the size of electrolyzers we can test in our collective laboratories is currently constrained to the 5 cm² area by equipment capabilities and resources. For example, scaling the system involves machining, leak-testing, and proofing the cathodic blocks, the anode cell, and the commercially available components (here, we used a Dioxide Materials'® zero-gap membrane-electrode assembly electrolyzer). Our 3D-printed components in this experiment are also close to the maximum size we can fabricate in our facilities without noticeable bending and curing deformations (we use Formlabs® UV-fixing resin, which is stable in alkaline solutions). Of course, we could fabricate larger-area electrodes using off-campus engineering firms. However, the cost and time to design and fabricate electrolyzers larger than 5 cm² electrode area will be more expensive, and we think that given this research's present state, the additional costs for larger electrodes and components are not yet justified. Larger-area

electrolysers certainly need to be tested, but that development and commercialisation work is beyond the scope of this manuscript.

Fortunately, we can extrapolate the results from our 5cm² electrode study using resistivity and circuit approximations to foresee how the NICC concept could be adapted for larger-area electrode systems. These insights will be necessary to design the experimental studies for larger-area electrolysers. For example, let's assume we compare a 2 cm x 2 cm electrode with area 4 cm² (whose busbars are the same as our experimental 2.25 cm x 2.25 cm electrodes) with a 100-fold larger electrode. This would be a 20 cm x 20 cm electrode with an area of 400 cm² (see below Fig. S21). By maintaining the same scaling factors as in our prototype reported in this manuscript and using a 2 cm x 2 cm electrode as the base case, we can expect the 100-fold larger electrode to have a busbar spacing of 20 mm, a busbar height of 100 μm, and a busbar width of 3 mm. The scaling principle we propose here is to scale the busbar dimensions based on the area increase of 100-fold, not the side length dimension increase of 10-fold. The approximate cross-sectional area of the busbars is then 0.3 mm². The current that travels through each busbar from the outside to the centre would then $I = (10 \text{ cm}) \times (2 \text{ cm}) \times (0.2 \text{ A cm}^{-2}) = 4 \text{ A}$ (for $j = 0.2 \text{ A cm}^{-2}$). Segmenting the 10 cm pathway into 20 parts (because current decreases along the length of the busbar), we can calculate the experienced voltage drop by the current to get to a certain point in the electrode using:

$$\Delta U_i = l \cdot I_i \cdot \frac{\rho}{A}$$

where l is the distance from the current collector, I the current passing through the segment, ρ the approximate resistivity of the copper busbar ($1.724 \times 10^{-6} \text{ } \Omega \text{ cm}$) and A the cross-sectional area of a busbar. The total voltage drop is then:

$$\Delta U_T = \sum \Delta U_i$$

As shown in Figure S21, the voltage drop from the exterior current collector to the centre of the small and large electrodes can then be maintained as ~80 mV with a reasonable busbar scaling.

Reducing busbar spacing and cross-hatching are added ways to reduce voltage drop and/or the required busbar diameters.

We have updated the text and the supplementary notes to reflect these calculations regarding the rationale for using busbars for larger electrode areas, as well as added the following Figure S21.

Supplementary Figure 21

- (a) Sketches for NICC designs on 4 cm² and 400 cm² electrodes with detailed NICC dimensions.
- (b) The maximum voltage drop along the busbars for a current density of 200 mA cm⁻² a 4 cm² electrode as detailed in (a), and (c) a 400 cm² electrode with a NICC design as in (a).

Added to pg. 16:

The demonstrations in this work were on a 2.25 cm by 2.25 cm (5 cm²) electrode. To examine the prospect of using the ePTFE and copper busbar approach for larger electrodes, we assessed the busbar dimensions required for a 20 cm by 20 cm (400 cm²) electrode to maintain an identical voltage drop from the exterior current collector to the centre of the electrode. These calculations are shown in the Supplementary Notes and in Supplementary Fig. 21 and indicate that the NICC approach can be applied to these larger electrodes using a busbar spacing of 20 mm, a busbar height of 100 µm, and a busbar width of 3 mm. These busbar dimensions could further be reduced using cross-hatched patterns or reduced busbar spacings.

“Looking at Fig. 4, since the bus bar is also copper, is it not also catalytically active? If so, I suggest re-evaluating the use of “non-interfering” since it would be involved in the reaction. Furthermore, since the copper on the busbar is significantly further (2 orders of magnitude) from the GDL than the Cu in between busbars, I request some explanation as to how the local chemical micro-environment differs between these catalytic sites of the electrode and how that is connected to the overall product distribution of the electrode. Sensitivity analysis experiments changing the height of the bus bar may help to assess the impact.”

We acknowledge this feedback that the term 'non-interfering' may confuse as it suggests the copper busbar is not participating in the electrochemical reactions. We have changed the terminology to 'non-invasive' with clarifications about its definition on pg. 13 of the revised document. We also performed additional experiments to confirm that excess copper in the busbars does not significantly impact selectivity or activity. In this response, we provide more context on the microenvironment and CO₂ availability for the 1 μm thick busbar, followed by a summary of the new control experiments to evaluate the busbar's activity and influence on observed performance. These findings are incorporated into the revised manuscript.

Firstly, on the microenvironment. Both our modelling results in Figure 1d and earlier reports in the literature (**Dinh et al.**, Science, 360, 783-787 (2018)) indicate that even at elevated current densities, excess CO₂ is available at 1 μm depths into the electrolyte, indicating that the busbars can be electrochemical active for CO₂ reduction. In Figure 1d we predict that at a current density of 300 mA cm⁻² there would be approximately 15 mM of dissolved CO₂ in the electrolyte at a distance of 1 μm from the cathode surface. In this sense, the busbars would likely be active for CO₂ electrolysis, and we have clarified this in the manuscript.

Second, we can assess the data in the original manuscript and data from the additional experiments to determine how the busbars influence the observed electrochemical activity for the 50 nm homogenous copper catalyst + NICC (labelled as 50 nm/NICC) compared to a control experiment using an electrode with a uniform 1 μm Cu layer on an ePTFE electrode (thus the Cu layer has the same depth as the busbars). The new data collected with the control experiment are included in the revised Table S4 and on the next page for convenience. Table S4 shows the selectivity differences between a 500 nm and a 1 μm electrode are small at most tested current densities, suggesting that the excess copper in the busbars does not significantly affect the selectivity or activity in the short-duration experiments. We observed more significant differences

at higher current densities, which may result from degradation, as discussed in the manuscript and later in these revisions.

In a second control experiment, we deposited 1 μm busbars with 0.3 mm widths on ePTFE without the 50 nm copper layer. Thus, most of the ePTFE is left exposed. These results are shown in Table S5 and Figure S20. At geometric current densities greater than 100 mA cm^{-2} , the ethylene selectivity of the ePTFE/NICC (without the 50 nm Cu layer) drops quickly as the small area of Cu supports a substantial current on its own. In fact, without the 50 nm copper layer, the busbar geometric area of $\sim 0.7425 \text{ cm}^2$ (0.3 mm width x 11 busbars x 22.5 mm length) must support the entire current, which at 100 mA cm^{-2} of total electrode areas is equivalent to a current density of 673 mA cm^{-2} through the busbars. CO_2 access is likely to be limited at these current densities due to a combination of CO_2 consumption and OH^- generation. From this data, we can conclude that although the copper busbar is electrochemically active, its primary function is as a current distributor when the busbar is in contact with a 50 nm copper layer.

We added a short paragraph in the manuscript to describe and discuss these these experiments and considerations of the busbar's electrochemical activity.

Table S4.

Selectivities for a 1 μm , 500 nm and 50 nm catalyst layers sputtered on ePTFE GDEs, and of a 200 nm Cu layer on a carbonous electrode (Sigracet 39BB). Data taken at increased current densities. FE's of liquid products are corrected and approximated to resting electrolyte volume at time of acquisition.

Faradaic Efficiency [%]

	j [mA cm ⁻²]	Methane [CH ₄]	Formate [HCOO ⁻]	Ethylene [C ₂ H ₄]	Ethanol [C ₂ H ₅ OH]	Acetate [CH ₃ COO ⁻]	Propanol [CH ₃ H ₇ OH]	Acetaldehyde [C ₂ H ₄ O]	C ₁	C ₂₊	H ₂
1 μm Cu	- 10	0.00	22.1	9.65	0.48	2.62	0.11	0.00	22.1	12.9	8.73
	- 50	0.00	10.8	33.5	13.0	0.57	6.83	0.00	10.8	53.9	8.42
	- 100	0.00	7.35	43.5	14.2	0.95	6.07	0.00	7.35	64.7	5.83
	- 150	0.00	7.92	46.5	25.3	1.38	9.22	7.85	7.92	90.3	4.45
	- 200	0.00	6.26	48.3	31.0	2.61	8.62	7.07	6.26	97.6	3.87
	- 300	0.00	5.13	47.9	32.3	3.27	6.80	8.51	5.29	98.8	4.87
500 nm Cu	- 10	1.11	16.0	17.8	15.0	7.40	0.13	0.14	17.1	40.5	16.1
	- 50	0.20	5.79	40.4	14.8	1.30	9.03	2.91	5.99	68.5	9.92
	- 100	0.30	4.70	45.5	23.5	2.19	6.79	3.58	5.00	81.5	6.50
	- 150	0.30	3.98	45.2	30.9	2.35	8.46	3.99	4.27	90.9	5.70
	- 200	0.33	3.59	45.8	33.5	2.97	6.97	3.63	3.91	92.8	6.05
	- 300	4.85	3.00	29.8	43.5	6.79	5.42	4.32	7.84	89.8	18.5

Table S5.

Selectivities for a uniform 50 nm Cu catalyst layers sputtered on an ePTFE GDEs, and of a 1 μm NICC pattern sputtered on a bare GDE. Data taken at increased current densities.

Faradaic Efficiency [%]

	j [mA cm ⁻²]	Methane [CH ₄]	Ethylene [C ₂ H ₄]	Hydrogen [H ₂]	Carbon Monoxide [CO]	Propane [C ₃ H ₈]	Gas products
Bare/NICC	- 10	0.00	34.1	4.15	20.3	0.50	59.1
	- 50	0.72	41.5	3.76	4.18	0.57	50.7
	- 100	6.23	24.9	25.2	0.86	0.12	57.4
	- 150	9.10	11.0	54.5	0.33	0.03	75.0
	- 200	5.09	5.42	69.6	0.26	0.02	80.4
	- 300	1.19	2.05	71.9	0.24	0.01	75.4
50 nm Cu	- 10	0.42	15.1	21.2	35.7	0.37	72.8
	- 50	4.64	40.7	6.83	8.93	0.60	61.7
	- 100	6.59	42.1	4.73	5.06	0.58	59.0
	- 150	13.3	38.2	6.42	1.65	0.38	60.0
	- 200	31.6	21.7	17.2	0.62	0.17	71.3
	- 300	29.4	2.46	30.8	0.09	0.03	62.8
50 nm Cu/NICC	- 10	0.39	20.4	8.98	25.7	0.36	55.8
	- 100	3.04	43.1	5.48	9.32	0.42	61.3
	- 150	3.26	45.4	4.47	6.45	0.77	60.4
	- 200	5.63	43.9	4.57	4.29	0.54	58.9
	- 300	27.5	18.8	20.54	6.59	0.10	73.6

Supplementary Figure S20: Comparative ethylene Faradaic efficiency for different electrodes including: (1) ePTFE and a 1 μm busbar layer (NICC), (2) ePTFE + 50 nm Cu layer, (3) ePTFE + 50 nm Cu layer + a 1 μm busbar layer (NICC)

Added to pg. 14 of the main text:

As the busbars themselves are made of copper, it is also interesting to examine if the electrochemical activity for the 50 nm/NICC case can be attributed to the 1 μm Cu busbars, as compared to the 50 nm Cu layer over the entire electrode. For instance, we model that sufficient dissolved CO_2 is expected to be present at 1 μm depths even at higher current densities (Fig. 1d). To understand this influence, we can compare the ethylene Faradaic efficiency for the 50 nm Cu case, the 50 nm/NICC Cu case, and an electrode only containing the 1 μm Cu busbars (e.g. ePTFE + NICC). As shown in Supplementary Table 5 and Supplementary Fig. 20 the 1 μm Cu busbars show the ethylene selectivity as a function of current density, while the combined 50 nm/NICC Cu case maintains ethylene selectivity with current density. We can then conclude that the greater performance of the 50 nm/NICC system over the 50 nm sample alone is not due to the electrochemical abilities of the 1 μm Cu busbars.

Minor comments:

1. “In the title, NICC = “non-invasive current collector”, but in the abstract, NICC = “non-interfering current collector”. It would be good to stay consistent with the acronyms that you define.”

We have corrected the naming in the abstract to ‘non-invasive.’

Reviewer #3's comments

“Catalyst detachment: The authors devote a section of their paper to demonstrate that the center of the 50 nm Cu ePTFE electrode experiences catalyst detachment because it is not under a sufficient reducing potential. If these catalyst sites are not being used for the reaction, the authors should explain how their detachment can be the cause of the electrode instability. The authors focus on the catalyst material which is not being used for the reaction. Could the active catalyst material (which is at a sufficiently reducing potential to protect the copper) be experiencing accelerated degradation since it is being overworked? Salt precipitation is another potential failure mechanism and it would be reasonable to assume that high local reaction rates drive up the pH and accelerate salt precipitation. This manuscript could benefit from further analysis or exploration to explain the cause of instability amongst the active catalyst sites. ”

We agree with the reviewer that two separate stability issues are likely at play. Firstly, in the thin films' centre region, the potential may be low enough that oxidation and dissolution occur, followed by redeposition in elevated potential regions. Second, at the electrode's perimeter, the chronopotentiometry setting's fixed current density will result in smaller geometric surface areas being forced to occupy much greater local current densities and, thus, greater potentials. These harsher conditions at the electrode's perimeter are expected to accelerate restructuring and morphology changes that steer selectivity towards hydrogen production and away from CO₂ reduction products.

We added greater detail in the manuscript to discuss these two possible degradation mechanisms, including the new illustration in Supplementary Figure S19.

Added to pg. 11:

On the other hand, selectivity figures for high current densities show a loss of selectivity and progressive deactivation of electrodes. The excessive equivalent current densities shown in Fig. S8 point to cathodic corrosion and restructuring as a primary deactivation mechanism at areas close to the current collector.^{52,53} An increased local potential on copper nanoparticles is known to result in accelerated restructuring of the surface, resulting in stabilization of certain intermediates and increase in partial current densities towards hydrogen.^{42,54}

All in all, it appears that the physical stability of copper electrodes is highly dependent on its conductivity and exposure to a certain potential value that ensures the catalyst remains in a fully reduced, metallic state. In relatively short time periods for the 50 nm sample areas that are

disconnected, or poorly connected, from the cathode current collector experience chemical oxidation and dissolution of the copper species. Conversely the areas that remain electrically connected near the perimeter of the electrode experience accelerated deterioration due to the increase of effective current densities (Fig. S19). This ultimately means that any solution that ambitions to stabilize copper electrocatalysts for CO₂RR must take the equalization of current distribution across the surface into account.

This is accompanied by a sketch in the SI to illustrate the described effects (Fig. S19).

Fig. S19.

Sketched deterioration mechanisms for sputtered copper catalyst layers on PTFE GDLs. Areas far from the current collector are progressively electrically isolated and experience chemical corrosion to soluble hydroxides. Areas close to the current collector experience cathodic corrosion resulting in restructuring of the catalyst nanoparticles.

Regarding salt accumulation: although it is true that at elevated pH, more CO₂ is buffered by the electrolyte to form carbonate ions, precipitation occurs only when the solubility product is

reached (Sassenburg et al., *ACS Energy Lett.* 2023, 8, 1, 321–331). For flow cells, where the catholyte is a mobile species being constantly refreshed close to the cathode, we expect lower salt deposition than that observed in membrane electrode assemblies (MEAs). While salt may block some active sites, we did not physically observe salt in any of our experiments.

“Stability benchmarking: The authors should provide a better explanation as to why their stability (only 4 hours) is so much less than other ePTFE electrodes demonstrations (>100 h). They hint that it is due to the lower electrolyte conductivity or higher current density (I suspect it is predominantly the current density at play), but I would suggest the authors run some control experiments demonstrating how stable their electrode could be if run in similar conditions to these other works. This could also assist in the discussion on my first concern above as they identify what is causing their electrodes to fail.”

We believe the main difference in the longevity of the electrodes in our study and other ePTFE reports in the literature is the active mass loading of the catalyst. For example, **García de Arquer et al.** (*Science*, 367 (6478), 661-666), reported a catalyst mass loading of around 2 mg cm⁻² of catalyst, which is a loading 67 times more Cu/cm² than the 0.03 mg cm⁻² Cu that we deposited in the 50 nm thick sputtered Cu films in our study (assuming a filling ratio of 0.66 in the sputtered film). Our 500 nm films have a mass loading of 0.30 mg cm⁻², and the 50 nm/NICC an average loading of 0.112 mg cm⁻². Our thickest Cu layer in the 200 nm/NICC film had a catalyst mass loading of 0.187 mg cm⁻², and as we show in the manuscript, this thickest Cu film was the most stable electrode in stability tests. We collected further evidence of the effect of Cu layer thickness on stability in a 7-hour stability test using a 1 μm thick Cu electrode (Fig. S25 below). Here, we see again that the total mass of Cu has a clear impact on stability, providing further evidence of the benefit of the Cu busbars on electrolyser stability. We have updated the text to include these reflections.

Added to pg. 14:

The current collection and distribution of activity over the catalytic area translates itself into an observable improvement of selectivity and stability for thin catalyst-layers. Comparing all three designs (500 nm, 50 nm and 50 nm/NICC on an ePTFE electrode) at constant potential during extended periods (4 hrs. max.), resulted in the ethylene selectivity displayed in Fig. 4c. Even though the onset selectivity of the NICC design is lower, a steep drop can be seen in non-NICC

electrodes. Whereas the NICC electrode displays a loss only 33% in ethylene selectivity, the 50 nm and 500 nm electrodes lose greater activity over the tested timeframes.

While these reported stability durations are lower than state-of-the-art reports,^{6,18} the mass loadings of copper in this work are orders of magnitude lower ($0.112 \text{ mg}_{\text{Cu}} \text{ cm}^{-2}$ for a 50nm Cu/NICC electrode versus $\sim 2 \text{ mg}_{\text{Cu}} \text{ cm}^{-2}$). This observation is supported by 7-hour constant potential measurements comparing (i) a $1 \mu\text{m}$ Cu electrode, (ii) a 200 nm Cu/NICC electrodes, and (iii) a 500 nm Cu sample (Supplementary Fig. 25) indicating higher mass loadings are an enabling factor in stability studies. Our results however still demonstrate the added stability towards C_2H_4 of the 50nm/NICC design versus the 500 nm and 50 nm counterparts (see Supplementary Table 6), though copper stability that plagues the CO_2 and CO reduction fields remains.

The following Fig. S25 was also added:

Fig. S25.

Long-term faradaic efficiencies towards ethylene and hydrogen of (a) $1 \mu\text{m}$ Cu ePTFE electrode, (b) 200 nm Cu/NICC electrode, and (c) comparison of stability towards ethylene at constant potential ($\sim -0.6 \text{ V vs. RHE}$).

“Non-intrusive nature of NICC: The authors claim carbon support layers used in previous electrode designs could evolve hydrogen at a reasonable operating voltage, so they instead they chose a copper busbar material which they claim is non-interfering. However, copper can also evolve hydrogen in the absence of CO₂ and the busbars are very thick (1000 nm) compared to the catalyst layers (50 or 500 nm). The current work does not persuade me that these copper busbars are non-intrusive. Have the authors done any modelling to predict what the CO₂ concentration is on these busbars deep in the electrolyte? Can they demonstrate experimentally how much (or how little) reaction current is taking place on the busbars? How would the voltage distribution and stability be impacted if another material was used (silver would be an interesting comparison since it is a poor catalyst for hydrogen evolution)?”

This comment from Reviewer 3 raises a similar comment to Reviewer 2, which we addressed above. We feel we already addressed this comment in the response to Reviewer 2’s comments through modelling, references to other modelling works, and our additional experiments.

The final remark about silver busbars is an interesting point we haven’t considered. Firstly, the conductivity of silver is about 5% better than copper, meaning the voltage drop would be 5% less. A silver busbar would also favour CO₂RR to carbon monoxide, which could be then be reduced on the copper catalyst to another product. Thus, a silver busbar would likely produce a different distribution of CO₂RR products than we observe with Cu busbars, possibly leading to increased selectivity for alcohols or acetate versus the pure copper case. Silver is also more electrochemically stable than copper, so silver busbars are likely more stable than copper.

On the other hand, *silver is typically 8 – 10 times more expensive than copper* (on a \$/mass basis) which adds significant cost to the electrolyser materials. We didn’t include any silver busbar experiments in the revised manuscript due to cost, time, and narrative reasons. However, we did add a short commentary about busbar metals in the Discussion section of the manuscript.

Added to pg. 16:

An interesting replacement for copper busbars in future studies would be the use of silver busbars as used in photovoltaics. Silver provides the best conductivity of any pure metal, with 5% greater conductivity than copper, which would reduce voltage losses along the busbar. Any by-product carbon monoxide formed during CO₂ reduction on the silver would also likely be utilized as a

reactant on the copper layers. Finally, silver is a more stable electrode material than copper, likely providing greater longevity. For these reasons future work should consider silver as a busbar material.

“1000 nm catalyst layer control experiment: The manuscript could benefit from a stability, activity distribution, and selectivity assessment of an ePTFE electrode with a 1000 nm catalyst layer (the same thickness of the busbars but applied uniformly across the electrode surface). The authors should share the polarization data for the 1000 nm, 50 nm, 500 nm, and 50 nm/NICC electrodes.”

We performed additional tests with a 1 μm Cu layer as discussed in our response to Reviewer 2's comments. The infrared thermographs of a 1 μm copper film electrode are included in the updated Figure 2b and 2c. As we expected, the new experimental data shows that the thicker Cu later has a more even activity distribution similar to the 500 nm Cu layer electrode. Further, the product mix of the 1 μm electrode was similar to 500 nm Cu layer, which is also expected due to the similarity of the local CO_2 concentration. As explored by others in CO_2 reduction literature, at lower current densities and short timescales the thickness of the catalyst layer typically does not have a significant effect on Faradaic efficiency because most of the CO_2RR activity happens near the gas-liquid interface (thus a thinner catalyst layer is preferred). Linear sweep voltammetry of the different catalysts is now provided in Fig. S24 with the 1 μm test data included in Table S4.

Table S4.

Selectivities for a 1 μm , 500 nm and 50 nm catalyst layers sputtered on ePTFE GDEs, and of a 200 nm Cu layer on a carbonous electrode (Sigracet 39BB). Data taken at increased current densities. FE's of liquid products are corrected and approximated to resting electrolyte volume at time of acquisition.

Faradaic Efficiency [%]

	j [mA cm ⁻²]	Methane [CH ₄]	Formate [HCOO ⁻]	Ethylene [C ₂ H ₄]	Ethanol [C ₂ H ₅ OH]	Acetate [CH ₃ COO ⁻]	Propanol [CH ₃ H ₇ OH]	Acetaldehyde [C ₂ H ₄ O]	C ₁	C ₂₊	H ₂
1 μm Cu	- 10	0.00	22.1	9.65	0.48	2.62	0.11	0.00	22.1	12.9	8.73
	- 50	0.00	10.8	33.5	13.0	0.57	6.83	0.00	10.8	53.9	8.42
	- 100	0.00	7.35	43.5	14.2	0.95	6.07	0.00	7.35	64.7	5.83
	- 150	0.00	7.92	46.5	25.3	1.38	9.22	7.85	7.92	90.3	4.45
	- 200	0.00	6.26	48.3	31.0	2.61	8.62	7.07	6.26	97.6	3.87
	- 300	0.00	5.13	47.9	32.3	3.27	6.80	8.51	5.29	98.8	4.87

500 nm Cu	- 10	1.11	16.0	17.8	15.0	7.40	0.13	0.14	17.1	40.5	16.1
	- 50	0.20	5.79	40.4	14.8	1.30	9.03	2.91	5.99	68.5	9.92
	- 100	0.30	4.70	45.5	23.5	2.19	6.79	3.58	5.00	81.5	6.50
	- 150	0.30	3.98	45.2	30.9	2.35	8.46	3.99	4.27	90.9	5.70
	- 200	0.33	3.59	45.8	33.5	2.97	6.97	3.63	3.91	92.8	6.05
	- 300	4.85	3.00	29.8	43.5	6.79	5.42	4.32	7.84	89.8	18.5
50 nm Cu	- 10	0.42	16.3	15.1	0.71	8.30	0.11	0.12	16.7	24.3	21.2
	- 50	4.64	5.08	40.7	14.9	2.57	5.59	6.39	9.72	70.1	6.83
	- 100	6.59	4.66	42.1	19.5	5.76	4.11	6.36	11.3	77.8	4.73
	- 150	13.3	3.99	38.2	24.4	7.90	4.49	6.47	17.3	81.5	6.42
	- 200	31.6	4.61	21.7	21.6	9.26	2.72	4.76	36.2	60.0	17.2
	- 300	29.4	4.90	2.46	13.4	5.12	2.83	3.63	34.3	27.5	30.8
200 nm Cu/C	- 10	0.15	30.8	2.59	11.8	6.41	0.11	0.12	31.0	19.8	28.6
	- 50	2.48	9.05	20.6	5.90	0.87	3.25	0.12	11.5	31.3	13.5
	- 100	4.15	8.79	32.3	17.2	2.32	4.86	3.59	12.9	61.2	9.13
	- 150	4.15	5.81	36.2	19.2	2.90	5.19	2.97	12.7	67.3	7.70
	- 200	5.13	5.99	37.2	25.6	4.25	4.76	4.06	11.1	76.5	7.98
	- 300	6.45	4.00	24.4	25.4	5.80	3.74	2.83	10.5	62.3	9.36

We have further added the polarization curves as requested as Figure S24 in the SI:

Fig. S24.

Polarization curves at -5 mV s^{-1} of $1 \mu\text{m}$, 500 nm , 50 nm and 50 nm Cu/NICC electrodes.

Minor comments:

1. *“Selectivity: The hydrogen selectivity for all experiments should be included (at least in the SI tables). Also, the product-specific breakdown of selectivity should be provided for the carbon paper electrode (Figure S1) in tabular form as it is for the other experiment sets.*
2. *Flooding of carbon electrodes: The authors should show the stability of ePTFE and carbon electrodes. They mention flooded carbon GDLs immediately in the first section but the paper would be stronger if they showed how fast flooding sets in.*
3. *Activity distribution of the 50 nm/NICC: The authors should indicate, graphically or in the caption, the orientation of the NICC (e.g., horizontal or vertical) as shown for Figure 4B.*
4. *PTFE vs. ePTFE: In the introduction, the authors should double check when PTFE and ePTFE are used. Here are some examples which I think should be referring to ePTFE: “...non-conducting PTFE GDE’s maintain hydrophobicity during sustained...”, “...electrolyzers using PTFE GDE’s have then shown the ability...”, and “...CO2 diffusion properties of PTFE GDE’s...”. Perhaps the authors should add a sentence clearly distinguishing PTFE and ePTFE.*
5. *Figure 1: I would suggest adjusting the colour scheme of Figure 1A to be more friendly towards colour-blind readers.*
6. *Typos: there are some minor typos in the manuscript (e.g., “...as is the case for the carbon electrode, the maximum local monoxide concentration...”).*

1. We have included hydrogen selectivities in Table S4, and expanded Table S6 with the CO faradaic efficiencies.
2. The rate of flooding of carbon electrodes is a function of several variables. Flooding sets in faster at higher liquid pressures (breakthrough of electrolyte) and faster at more cathodic potentials. Our lab published a work on the rate of flooding for several catalysts sputtered on carbon GDEs (K. Yang et al., **ACS Energy Lett.** 2021, 6, 1, 33–40). Herein we showed that flooding occurs on a bare GDL after only 30 minutes at current densities as low as $\sim 7 \text{ mA cm}^{-2}$. Relative to a stable operating system, this onset is fast, so it is safe to assume a carbonous electrode will have a flooded MPL for most of its operating lifetime.
3. We have included x-y and x-y-z axes in Figures 2 and 4 sketches and explicitly indicated the orientation in Figures 4a and 4b.
4. We have corrected the text in the introduction for further clarity. The introduction of the material in page 2 now reads as follows:

“To avoid flooding issues during long-term CO₂RR operation, researchers have aimed to increase the hydrophobicity of gas-diffusion layers (Fig. 1a).¹⁰ One successful approach to do so is using super-hydrophobic expanded polytetrafluoroethylene (PTFE) gas-diffusion layers.^{11–15} These GDLs consist of an expanded PTFE (ePTFE) micro-porous layer (MPL) supported by a coarser polymeric layer (e.g. polyethylene, PE) acting as the macro-porous layer. [...]”

5. We have changed the green-red gradients in Fig. 1a by the manuscript-wide palette of blue and red.
6. We have revised the text for typos and have corrected the suggested sentence.

Reviewer #4's comments

“The operando IR tomography is a valuable technique for evaluating the current distributions, however, it does not provide solid evidence on the factors that lead to that distribution over the electrode surface.”

We also acknowledge that IR thermography does not directly observe current distribution across the electrode area. Instead, we describe that this IR tomography method can be used as an indirect measurement that can be used with other measurements and characterisations to infer the distribution of current densities for various electrode scenarios. Based on this reviewer's comment, we have tried to clarify the limitations of the IR tomography method more explicitly in the manuscript's introduction (pg 7, revised text below). Specifically, we comment that the poor conductivity of the 50 nm layers versus thicker 500 nm layers will lead to substantial voltage variations on the electrodes, leading to different current distributions.

To further test these hypotheses, we conducted 2-electrode probes to measure the ohmic drop in electrodes with 1 μm Cu-films and 50 nm Cu-films over defined distances without busbars. The data from these measurements are included in Table S3 and illustrate the comparative resistance of the sample layers, which would lead to current distributions of potentials to the electrode perimeter.

Table S3.

2-electrode probe resistances measured for a 50 nm and a 1 μm Cu layers.

Cu layer thickness	$R_{\text{path},1\text{cm}}$ [$\text{m}\Omega \text{ cm}$]
50 nm	1.47
1 μm	0.17

Added to pg. 7:

“Then, using our IR thermography set-up (Fig. 2a, Figs. S3-S6), we characterized the evolution of temperature distribution during a linear polarization of the cathode. Infrared thermography is a suitable method to study the local activity on thin GDEs, with which one can relate the increase in temperature of the backbone to local current density on the catalyst of the electrode.²⁴ While it is

not a direct *in situ* methodology that directly explains current distributions, it is a powerful proxy to study current distributions of GDEs under differing catalyst layers (see the Supplementary Text).”

“The interpretation of the differences in activity between the carbon GDL and the ePTFE is very simplistic. As shown in figure S15 and S16, the morphology of the obtained Cu films is very different and by itself can lead to different activities and product distributions.”

We agree that the differences in selectivity for the ePTFE and carbon-based gas-diffusion layers are complicated by the differences in the thickness, morphology and surface area of these two types of GDES. As our primary emphasis is the comparative performance over time due to flooding, we have greatly simplified this portion of the manuscript as shown below:

Changed text on page 5/6:

To illustrate the effect of a flooded MPL, we employed a simple reaction-diffusion model at steady-state for both architectures. Solving the model for current densities of 300 mA cm^{-2} along the gas-liquid phase-boundary, we see that the concentration of dissolved CO_2 is severely hampered in the flooded carbon MPL case compared to an ePTFE electrode (Fig. 1d and e, see Supplementary Information for details on the model’s parameters). ~~As the local concentration of dissolved CO_2 and CO_2 : CO ratio have been shown to be determinant in the C_{2+} selectivity of catalysts on a GDE,^{27–29} flooding of GDEs must play a considerable role in the selectivity and stability of these architectures.~~

Comparing the product selectivity of a carbon-based electrode (Sigracet® 38BB) and an ePTFE electrode (Sterlitech® Aspire QL822) over time highlights the limitations of the carbon support ~~which is more susceptible to flooding due to electrowetting of the carbon support.~~ Here we used a three-chamber flow cell using 1M KOH as both the catholyte and anolyte. The carbon-based GDE was sputtered with a 200 nm Cu layer, and the ePTFE electrode with a 500 nm one. ~~For these two different electrodes we then ran constant current densities experiments from -10 mA cm^{-2} to -300 mA cm^{-2} .~~ Here we observed that the ePTFE electrode exhibited a superior selectivity towards hydrocarbon C_{2+} products (ethylene, ethanol, acetate, propane and propanol) across the board (Fig. S1). ~~Carbon monoxide (CO), a known intermediate to these~~

hydrocarbons, is noticeably lower for these electrodes, which points to a higher utilization rate of the locally generated CO.^{30,31} The carbon-based electrode showed higher selectivity to CO, relative to hydrocarbons selectivity at current densities below 50 mA cm^{-2} , and had a clearly lower selectivity towards higher hydrocarbons at industrially relevant current densities ($\sim 65\%$ vs. $\sim 80\%$ at 200 mA cm^{-2}) than the ePTFE electrode. We can then conclude the product compositions of each electrode architecture respond, at least partially, to the effect these have on the concentrations of species around the catalyst layer. For example: carbon monoxide is known to have a considerably lower solubility in water than CO_2 .³² For a fully flooded MPL, as is the case for the carbon electrode, the maximum local carbon monoxide concentration is then greatly reduced compared to the ePTFE electrode. While the copper catalyst's on each electrode support are not identical in morphology or surface area, the disparity in selectivities towards the higher value C_{2+} products show the influence architecture design can have on the performance of a GDE.

“As the authors also analyze, there are significant morphological modifications on the electrode surface during reaction (see the relative difference in the capacitance of the electrodes in figure 4, for example). The rate of the surface corrosion can also be the difference between the electrodes with different Cu thicknesses.”

Surface corrosion and reconstruction could explain the effects and be a positive feedback mechanism as degradation occurs. Looking closely at Figure 4d, we can also see different deterioration rates. While for the 50 nm Cu layer the increase in capacitance is 10-fold (0.14 mF to 1.7 mF), the rate for the 50 nm/NICC is 4-fold (2 mF to 10 mF). For the 500 nm Cu we observed a rate of around 2x (10 mF to 19.5 mF). If corrosion alone were the main driver of decay in the performance of these electrodes, we would expect the deterioration rate for the 500 nm layer to be bigger than that of the NICC design. Instead, we argue corrosion and morphological modifications play a role in the deactivation. For example by losing active area, the imposed total current must then be spread over a smaller active area, increasing the local current density. This drives the overpotential of the cathode to more cathodic values over time as the geometric area with catalyst becomes smaller, and by extension impacts CO_2 availability and selectivity (see Reviewer 2 comment 2 for a separate but related discussion).

We have updated the last paragraph before the discussion to reflect this observation better (addition in yellow):

Added to pg. 14:

“A way of monitoring the evolution of surface roughness during electrolysis is the measuring of double-layer capacitance in the used GDEs.⁶⁰ To do so, we performed cyclic voltammetry before and after long electrolysis runs (~4 h) between -0.3 and 0.3 V vs. Ag/AgCl. After measuring charging currents and linearizing against scan-rate (see Methods), the capacitance of the NICC design shows a noticeable increase against its 50 nm counterpart (Fig. 4d). The thicker, 500 nm electrode, however, still displays a bigger capacitance overall. The change in capacitance is relatively smaller for the 50 nm/NICC and 500 nm designs than for the 50 nm one, indicating a dramatic increase in surface roughness. Compared to the 50 nm/NICC design, the 500 nm layer displays a lower capacitance change (around two-fold, 10 mF before and 19.5 mF after). This indicates that, while corrosion of the electrode can influence selectivity directly, it is not the main driver of selectivity changes, as the 50 nm/NICC shows an improved selectivity towards ethylene over time. While the sharp disparity between before and after measurements can also be due to the presence of oxide species in the fresh samples tested before electrolysis, the noticeable increase for the 50nm/NICC sample suggests a considerable larger area of the electrode is electrically connected. The improved current density distribution and more equal overpotential distribution, then, appear to also influence degradation of these electrodes directly.”

REVIEWERS' COMMENTS

Reviewer #2 (Remarks to the Author):

The authors have sufficiently addressed my feedback from the first round of peer review. I now recommend this work for publication.

Reviewer #3 (Remarks to the Author):

The revised manuscript addresses my concerns and I now recommend it for publication.

Reviewer #4 (Remarks to the Author):

The authors have improved the manuscript significantly and addressed appropriately all the comments raised by the referees. The manuscript can now be accepted for publication at Nature Communications.